# Multifunctional Biomass-Based Ionic Liquids/CuCl-Catalyzed CO_2_-Promoted Hydration of Propargylic Alcohols: A Green Synthesis of α-Hydroxy Ketones

**DOI:** 10.3390/ijms25031937

**Published:** 2024-02-05

**Authors:** Ye Yuan, Siqi Zhang, Kang Duan, Yong Xu, Kaixuan Guo, Cheng Chen, Somboon Chaemchuen, Dongfeng Cao, Francis Verpoort

**Affiliations:** 1State Key Laboratory of Advanced Technology for Materials Synthesis and Processing, Wuhan University of Technology, Wuhan 430070, China; fyyuanye@whut.edu.cn (Y.Y.); chengchen@whut.edu.cn (C.C.); sama_che@hotmail.com (S.C.); cao_dongf@whut.edu.cn (D.C.); 2School of Material Science and Engineering, Wuhan University of Technology, Wuhan 430070, China; 317676@whut.edu.cn (S.Z.); 317394@whut.edu.cn (K.D.); whuthccs@163.com (Y.X.); 331282@whut.edu.cn (K.G.); 3Research School of Chemical and Biomedical Technologies, National Research Tomsk Polytechnic University, Lenin Avenue 30, 634050 Tomsk, Russia

**Keywords:** ionic liquid, CO_2_, biomass, catalysis, α-hydroxy ketones

## Abstract

α-Hydroxy ketones are a class of vital organic skeletons that generally exist in a variety of natural products and high-value chemicals. However, the traditional synthetic route for their production involves toxic Hg salts and corrosive H_2_SO_4_ as catalysts, resulting in harsh conditions and the undesired side reaction of Meyer–Schuster rearrangement. In this study, CO_2_-promoted hydration of propargylic alcohols was achieved for the synthesis of various α-hydroxy ketones. Notably, this process was catalyzed using an environmentally friendly and cost-effective biomass-based ionic liquids/CuCl system, which effectively eliminated the side reaction. The ionic liquids utilized in this system are derived from natural biomass materials, which exhibited recyclability and catalytic activity under 1 bar of CO_2_ pressure without volatile organic solvents or additives. Evaluation of the green metrics revealed the superiority of this CuCl/ionic liquid system in terms of environmental sustainability. Further mechanistic investigation attributed the excellent performance to the ionic liquid component, which exhibited multifunctionality in activating substrates, CO_2_ and the Cu component.

## 1. Introduction

Nowadays, the excessive emission of CO_2_ has led to a range of environmental and social problems such as global warming, glaciers melting and sea levels rising [1,2,3,4]. As a result, the effective and efficient management of CO_2_ has become an urgent concern for scientists and engineers. From the perspective of synthetic chemistry, CO_2_ is regarded as an ideal substitute for traditional phosgene and carbon monoxide [5,6,7,8] due to its wide availability, low cost, easy accessibility, non-toxicity and environmental friendliness [9]. This has prompted the exploration of CO_2_ for the production of fine chemicals, as it holds the potential to not only create economic value but also mitigate the greenhouse effect [10]. However, due to the thermodynamic stability and kinetic inertness of CO_2_, its effective activation still remains a significant challenge. Consequently, the exploration of effective catalysts for CO_2_ activation and the designation of feasible reaction routes for CO_2_ conversion have accordingly emerged as critical focal points in the pursuit of CO_2_ utilization.

In recent years, considerable progress has been achieved in this area [11,12,13,14,15,16,17,18,19], including effective routes for utilizing CO_2_ in the production of methyl urea, urea, salicylic acid, organic carbonate, methanol, polycarbonate, etc. Among them, α-hydroxy ketones are crucial organic skeletons that generally exist in a variety of natural products and are frequently used as synthetic precursors for high-value chemicals [20,21]. The hydration of propargyl alcohols was an ideal method for the production of α-hydroxy ketones due to its 100% atom economy and the accessibility of diverse starting materials. However, the direct hydration of propargyl alcohols typically required catalysts involving strong acids like H_2_SO_4_ [22,23] and rare, toxic metal salts such as Au [24,25,26,27,28,29,30,31,32,33,34], Ag [35,36,37,38,39,40] and Ru [41], which resulted in harsh conditions and the undesired side reaction of the Meyer–Schuster rearrangement [42]. Based on this, the indirect hydration of propargyl alcohols has emerged, in which the cyclization of CO_2_ and propargyl alcohols firstly occurred to obtain the α-alkylidene cyclic carbonates, followed by an in situ hydration of these carbonates to give the desired α-hydroxy ketones [43]. This CO_2_-promoted indirect hydration generally proceeded under basic conditions, thus eliminating the Meyer–Schuster rearrangement in essence. Moreover, the reaction conditions were relatively milder than those for the direct process and thus have attracted great attention from researchers and engineers. 

Over the past decade, numerous catalysts have been investigated for this CO_2_-promoted indirect hydration. In 2014, Jiang et al. introduced a AgOAc/1,8-diazabicyclo [5.4.0]undec-7-ene (DBU) system, which successfully catalyzed this CO_2_-promoted hydration of internal and terminal propargyl alcohols, yielding the target products in high yield [44]. However, this system necessitated a high loading of Ag salts, traditional volatile solvents and strong bases. Moreover, its working pressure reached 20 bar of CO_2_. In 2015, Liu et al. presented a range of task-specific ionic liquids (ILs) for this hydration, operating effectively under 1 to 10 bar of pressure and offering a recycling and reusing ability [45]. In 2019, He et al. developed a Cu_2_O/DBU catalytic system composed of Cu_2_O (20 mol%), DBU (50 mol%) and phosphine ligands (20 mol%), which efficiently converted various types of terminal propargyl alcohols into the corresponding target products in CH_3_CN under 1 bar of CO_2_ [46]. In 2020, Yuan et al. developed a AgOAc/ILs system for this reaction, which operated under atmospheric CO_2_ pressure and solvent-free conditions to produce α-hydroxy ketones [47]. Furthermore, they established a Zn-based catalytic system that exhibited excellent catalytic activity for the target reaction under simulated flue gases, with the Zn species generated from pigment wastes [48]. More recently, two heterogeneous catalysts, namely a silver-anchored porous aromatic framework catalyst (Ag@PAF-DAB) and an amino-functionalized organic polymer Cu@Co-PIL-N4 loaded with highly dispersed CuI, have also been reported [49,50].

In the context of the aforementioned work, IL-involved systems have emerged as pivotal roles. ILs are composed of anions and cations with melting points below 100 °C, which have been widely applied in CO_2_ capture and catalytic conversion due to their advantages of designability, stability and catalytic activity [51]. However, the utilization of ILs for the CO_2_-promoted hydration of propargyl alcohols remains relatively uncommon and costly to date. Furthermore, some IL catalysts still required elevated CO_2_ pressure to reach high activity, which largely limited their further applications.

Biomass compounds are natural organic substances such as cellulose, wood chips and fructose. Nowadays, these materials have been utilized in numerous emerging areas due to their greenness, abundance, renewability, etc. [52,53]. These compounds can be further disassembled into biomass-based platform compounds [54,55], such as the 12 biomass molecules of succinic acid, 2,5-furan dicarboxylic acid, 3-hydroxypropionic acid, itaconic acid and levulinic acid, proposed by the US Department of Energy [56]. These derived compounds typically contain carboxylic or hydroxyl groups, indicating their great potential for ionization and application as the anions of ILs. Importantly, ILs containing carboxylic or hydroxyl ions have been proven to exhibit significant interaction with CO_2_ molecules [57]. As a result, the biomass-based ILs have the potential to demonstrate catalytic activity for CO_2_ capture and activation. In addition, these ILs are derived from biomass, which indicates their economical, renewable and eco-friendly nature, aligning with the requirements of modern green and sustainable development.

Herein, a series of biomass-based ILs were designed and synthesized, which were further combined with the economical CuCl for the catalysis of the CO_2_-promoted hydration reaction of propargyl alcohols. Particularly, this catalytic system worked under 0.1 MPa of CO_2_ with a low metal loading of CuCl (1 mol%) without any traditional organic solvents or ligands. Further evaluation of the green metrics revealed the superiority of this CuCl/IL system in terms of environmental sustainability.

## 2. Results

In this section, the catalytic performance of various biomass-based ILs for the CO_2_-promoted reaction was investigated, with the employment of 2-methyl-3-butyn-2-ol (**1a**) as the initial substrate (Table 1). The screening of ILs commenced with the blank experiments, demonstrating that the reaction could not proceed without catalysts (entry 1). Subsequently, it was observed that neither the metal salts nor the ILs alone could catalyze the target reaction (entries 2, 3). Notably, the absence of CO_2_ hindered the target reaction, indicating the crucial role of CO_2_ (entry 4). After the blank experiments, the investigation of the catalytic activity of different ILs was performed with the metal salt component fixed as CuCl. The screening was initially focused on the optimization of various anions (Figure 1), derived from levulinic acid (Lev), lactic acid (La), itaconic acid (ITa) and succinic acid (Sa). Experimental results revealed that [Lev] obtained the highest yield (entries 5–8), which was consequently identified as the optimal anion. Subsequently, the effects of cations (Figure 1) on the activity of ILs were explored, with the catalytic performance order revealed as [C_2_C_1_im] > [N_4444_] > [P_4444_] > [C_4_C_1_im] > [DBUH] > [DBNH] (entries 5, 9–13). The slight difference between [C_2_C_1_im] and [C_4_C_1_im] cations in catalytic activity can be attributed to the physical properties of the corresponding ILs. Generally, [C_4_C_1_im][Lev] exhibited higher viscosity than [C_2_C_1_im][Lev], resulting in a thicker reaction system that was more difficult to blend and stir. Consequently, the best choice of ILs was identified as [C_2_C_1_im][Lev]. With the optimal IL in hand, the metal salts in the catalytic system were further explored, which was mainly focused on economical Cu salts due to their inherent affinity to triple bonds, such as CuCl, CuBr, CuI, Cu_2_O, Cu_2_S, CuCl_2_, Cu(OAc)_2_, and CuSO_4_ (entries 13–20). It was found that both Cu (I) and Cu (II) salts exhibited considerable catalytic activity towards the target reaction, with CuCl achieving the highest yield of 95% (entry 13). Consequently, the optimal catalytic system was determined as CuCl/[ C_2_C_1_im][Lev].

After identifying the optimal catalytic system as CuCl/[C_2_C_1_im][Lev], the reaction conditions were subsequently explored (Table 2). The amount of [C_2_C_1_im][Lev] was gradually increased from 0.5 to 1 equiv., resulting in the highest yield of 95% (entries 1–3). A similar trend was observed for the amount of CuCl, with the yields increasing as the amount of CuCl rose from 0.25 to 1 mol%. (entries 3–6). Subsequent investigations focused on the reaction temperature. An increase in reaction temperature from 40 to 80 °C significantly improved the catalytic yield from 5% to 95%. However, at a higher temperature of 100 °C, the reaction could not proceed further, leading to the identification of 80 °C as the optimal temperature (entries 3, 7–9). Furthermore, the influence of reaction time was investigated, and the yield increased gradually as the reaction time extended (entries 3, 10–12). At a reaction time of 12 h, the yield of the product reached 95% (entry 3). Since the target reaction proceeded smoothly under 1 bar of CO_2_, experiments for higher pressure were not performed. Consequently, the final reaction conditions were determined as CuCl (1 mol%), [C_2_C_1_im][Lev] (1 equiv.), 80 °C, CO_2_ (0.1 MPa) and 12 h.

After determining the reaction conditions, the substrate scope of the CuCl/[C_2_C_1_im][Lev] system was explored (Table 3). The experimental results revealed that the majority of tertiary propargylic alcohols with various substituents effectively produced the corresponding products (**1a**–**1g**). Notably, the steric effects of the substituents significantly influenced the reactivity of substrates during the formation process of α-hydroxy ketones. Substrates with less sterically hindered substituted groups such as methyl and ethyl provided high yields of 85–95% within 12 h. However, for the propargylic alcohol **1g** containing a bulky phenyl group, the yield was only 33% under the same conditions, which could be improved via extending the reaction time. Additionally, attempts were made to react primary and secondary propargylic alcohols (**1h**, **1i**), but the corresponding products could not be produced. This might be attributed to the lack of gem-dialkyl effects in these substrates, resulting in the failure of cyclizing CO_2_ and propargylic alcohol [58,59]. Subsequently, the recyclability of the CuCl/[C_2_C_1_im][Lev] system was investigated. After being recycled and reused three times, the catalytic system could still catalyze the target reaction to produce the α-hydroxy ketones with a yield of 90%, demonstrating its considerable stability and recyclability.

In addition to substrate scope and recyclability, the greenness of the reaction process is another important aspect of modern sustainable development, which can be quantified by utilizing the “green metrics”, including atom economy (AE), E-factor, carbon efficiency (CE), reaction mass efficiency (RME), mass intensity (MI) and mass productivity (MP) [60]. These well-defined and objective metrics provide quantitative standards for the greenness evaluation (Part 2, Appendix A). In this context, the green metrics of the CuCl/[C_2_C_1_im][Lev] system and the calculable systems reported by other researchers were compared, based on the hydration of **1a** (Table 4). Upon evaluating a total of six green metrics, the CuCl/[C_2_C_1_im][Lev] system provided superior values in four aspects (AE, E-factor, MI, MP). This result indicated that the CuCl/[C_2_C_1_im][Lev]-catalyzed hydration of propargyl alcohols was a relatively greener process.

## 3. Discussion

### 3.1. Identification of Tandem Mechanism

Based on previous reports [47], the CO_2_-promoted hydration of propargyl alcohols may proceed via a two-step tandem reaction mechanism. Substrates and CO_2_ may first undergo cyclization to form α-alkylidene cyclic carbonates, followed by the in situ hydration of these carbonates with the release of CO_2_ during the process. To investigate whether the CuCl/[C_2_C_1_im][Lev]-catalyzed reaction aligns with this proposed mechanism, a series of control experiments were conducted. Initially, **1a**, CO_2_ and CuCl/[C_2_C_1_im][Lev] were introduced into the system under the optimal reaction conditions, with the omission of H_2_O. After 12 h, the reaction was terminated, and the sample was analyzed using ^1^H NMR. Comparing the spectrum of the reaction mixture with that of pure α-alkylidene cyclic carbonate, the characteristic peaks of α-alkylidene cyclic carbonates appeared on the spectrum of the reaction mixture (Figure 2a), indicating the generation of these carbonates in this process. Subsequently, the pure α-alkylidene cyclic carbonates were added with H_2_O and allowed to react for 12 h under the catalysis of [C_2_C_1_im][Lev]. Upon completion of the reaction, the sample was analyzed using ^1^H NMR. The result showed that the desired α-hydroxy ketones were successfully obtained (Figure 2b). These experiments demonstrated that the CuCl/[C_2_C_1_im][Lev]-catalyzed CO_2_-promoted hydration of propargyl alcohols followed the proposed tandem reaction mechanism, and the ILs act as the pivotal catalyst for the hydration process.

### 3.2. Activation of the Propargylic Alcohols

Upon identifying the tandem mechanism of the target reaction, the first cyclization step was further studied. In this step, the activation of the hydroxyl group in propargyl alcohols was quite crucial, which would initiate the whole catalytic process. Therefore, the investigation focused on identifying the component responsible for this crucial activation. Typically, this activation could be indicated via the shape and chemical shift of the hydroxyl signal peak in the ^1^H NMR spectrum. For pure **1a**, the hydroxyl proton exhibited a distinct characteristic peak at δ = 5.29 ppm (Figure 3a), indicating its inactivated state. However, upon the addition of DBU, a well-established organic base known for its effective hydroxyl group activation, the signal transformed into a broad peak with a different chemical shift (Figure 3b). This represented the activated state of the hydroxyl proton. Subsequently, the two components of the catalytic system were successively scrutinized. The addition of CuCl to **1a** did not produce any discernible change in the ^1^H NMR results (Figure 3c), suggesting that CuCl was not capable of activating **1a**. Conversely, the combination of [C_2_C_1_im][Lev] and **1a** led to a broad and shifted hydroxyl peak similar to that in the DBU/**1a** system (Figure 3d). This result implied that [C_2_C_1_im][Lev] played a pivotal role in the activation of the hydroxyl group in **1a**.

### 3.3. Generation of NHC-CO_2_ Adducts

It has been reported that under basic conditions, imidazole compounds might interact with CO_2_, resulting in the generation of free N-heterocyclic carbenes (NHCs) that are capable of CO_2_ capture and activation via the formation of NHC-CO_2_ adducts (Figure 1) [62,63]. In the aforementioned investigations, [C_2_C_1_im][Lev] was identified to provide suitable basic conditions for the activation of the hydroxyl group. Therefore, subsequent exploration focused on whether this imidazole IL could produce NHC-CO_2_ adducts. In this study, CO_2_ was introduced into [C_2_C_1_im][Lev], and the mixture was allowed to stir for 12 h. It could be observed that the solution became turbid gradually. Upon completion, the solution was analyzed using ^13^C NMR. In the spectrum, a new signal peak at δ = 154.60 ppm was observed (Figure 4a), which was consistent with the characteristic peak of CO_2_ adducts reported in the literature [63], indicating the successful formation of NHC-CO_2_ adducts.

### 3.4. Interaction between [C_2_C_1_im][Lev] and Cu Species

Apart from the function in activating hydroxyl groups and generating NHC-CO_2_ adducts, the [C_2_C_1_im][Lev] component was further identified to improve the catalytic activity of CuCl via the interaction between [Cu] and carbonyl groups in the [Lev] anions. In this context, an analogous IL to [C_2_C_1_im][Lev] was synthesized, differing only in the absence of a carbonyl group in its anion structure (Figure 2, entry 1). Subsequent experiments were designed to compare the catalytic performance of these two ILs. The results illustrated a significant decrease in catalytic yields in the absence of carbonyl groups, demonstrating the indispensable roles of carbonyls in ILs. Further evidence of the interaction between the carbonyl groups and [Cu] was revealed via the ^1^H NMR of the CuCl/[C_2_C_1_im][Lev] mixture (Figure 5). At 80 °C, significant interactions were observed as the signals of CH_2_ groups adjacent to the carboxyl groups (dashed part, Figure 5) shifted and broadened. This contrasts with the sharp triplet peaks observed at 25 °C, suggesting that the interaction at lower temperature was less significant. This observation provides a plausible explanation that the CuCl/[C_2_C_1_im][Lev] system exhibited better activity at a relatively higher temperature.

### 3.5. Proposed Mechanism for the Catalytic Process

Based on the aforementioned experiments and discussions [44,57,64,65,66], the following mechanism for the CO_2_-promoted hydration of propargylic alcohols catalyzed via the CuCl/[C_2_C_1_im][Lev] system was proposed, which could be outlined in two steps (Figure 3). Firstly, [Lev] activates the hydroxy group of **1a**, thereby enhancing the nucleophilicity of the hydroxy oxygen and facilitating its subsequent attack on the CO_2_ molecule. Simultaneously, the Cu species coordinates with the unsaturated triple bond of **1a**, leading to the formation of the transition state (**TS**). In the next stage, the negative hydroxyl oxygen atom bonds with the positive carbon center of CO_2_, incorporating the inert CO_2_ molecule into the organic skeletons, resulting in the formation of intermediate **I**. Subsequently, the negative oxygen of the CO_2_ moiety continues to attack the triple bond, activated via the Cu species, leading to intramolecular cyclization and the formation of intermediate **II**. Finally, the proton is transferred back to the organic skeletons, resulting in the generation of the key α-alkylidene carbonate (intermediate **III**). In the second step, H_2_O initiates a nucleophilic attack on intermediate **III** with the catalysis of the basic [Lev]. This leads to the ring-opening reaction and the generation of intermediate **IV**. Afterwards, this intermediate undergoes keto–enol isomerization, subsequently releasing the CO_2_ molecule and yielding the final target product **2a**.

## 4. Materials and Methods

The series of biomass-based ILs used in the experiments were synthesized according to the reported literature (Part 1, Appendix A) [57,67,68]. Unless otherwise specified, all the propargyl alcohol substrates (98%) and biomass acids (95%) used in the experiments were purchased from Aladdin (Shanghai, China), TCI (Tokyo, Japan), Sigma-Aldrich (Shanghai, China), Macklin (Shanghai, China), Alfa (Shanghai, China), etc. and directly used without further purification and drying. The purity of the CO_2_ used for purging and reacting was 99.9%, supplied by Wuhan Xiangyun Industry and Trade Co., Ltd., (Wuhan, China).

The ^1^H NMR spectra were recorded on a Bruker Avance III HD 500 MHz spectrometer, with the internal standard TMS (δ = 0 ppm) serving as the reference. Meanwhile, the ^13^C NMR spectra were recorded at 126 MHz in CDCl_3_ (δ = 77.23 ppm) or DMSO-*d*_6_ (δ = 39.50 ppm), with the solvent peaks as the internal references. The data were given as chemical shifts (ppm) and coupling constants (Hz), respectively.

### 4.1. The CO_2_-Promoted Hydration of Propargylic Alcohols

CuCl (0.025 mmol), 1-ethyl-3-methylimidazolium levulinic ([C_2_C_1_im][Lev], 2.5 mmol), propargylic alcohols (2.5 mmol) and H_2_O (5 mmol) were added into a Schlenk tube. Subsequently, the system was purged three times with CO_2_ and then stirred at 80 °C under 0.1 MPa of CO_2_ for the required time. When the reaction was completed, the mixture was extracted with diethyl ether (3 × 15 mL). The upper organic phases were concentrated under a vacuum to give the crude products, which were further purified via column chromatography on silica gel using petroleum ether/ethyl acetate (*v*/*v*, 100:1–20:1) as the eluent.

### 4.2. Procedures for Recycling the Catalytic System

After the reaction was completed, the mixture was extracted three times with diethyl ether (3 × 15 mL). The lower layer was then dried under a vacuum for 4 h to totally remove the residual solvents, reactants and products. After drying, the catalytic system could be reused for the next round.

## 5. Conclusions

In summary, a CuCl/[C_2_C_1_im][Lev] system was developed for the CO_2_-promoted hydration of propargylic alcohols under 1 bar of CO_2_ pressure without the use of organic volatile solvents or additives. The system demonstrated high activity, a wide substrate scope and significant recyclability. Notably, the catalytic process was identified as superior in terms of green metric evaluation. A comprehensive mechanistic investigation revealed that the exceptional performance of this system can be attributed to the triple function of the [C_2_C_1_im][Lev] component, which encompasses the activation of substrates, generation of NHC-CO_2_ adducts and interaction with the Cu component.

## Data Availability

Data is contained within the article and Appendix A.

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
