# Peer review of "Multifunctional Biomass-Based Ionic Liquids/CuCl-Catalyzed CO2-Promoted Hydration of Propargylic Alcohols: A Green Synthesis of α-Hydroxy Ketones"

_ijms, 2024, doi:10.3390/ijms25031937_

Round 1

Reviewer 1 Report

Comments and Suggestions for Authors

The manuscript studied the performance of biomased-based ILs as the catalysts for the CO2-promoted hydration of propargylic alcohols. The manuscript is well prepared with supportive data validations.

Please consider the following comments:

- It is acceptable to label the ILs in this study as "biomass-based ILs." However, using the acronym BILs consistently throughout the manuscript may be unnecessary since the individual compounds are pure. The use of the BILs acronym would be more justified if the authors separated the compounds directly from the biomass and converted them into BILs.

- Line 77: "As a new type of green and environmentally friendly medium, ..." it is still debatable to classify ILs as green and environmentally friendly. 

- Table 1: the table should appear immediately after the table. Also, For entry 1-4, the symbol "/" does not seem appropriate to indicate the yield, metal salt and IL.

Author Response

We appreciate the comments and suggestions of the reviewer. All the responses and modifications are showed in the attachment. 

Reviewer 2 Report

Comments and Suggestions for Authors

The authors have recently published two articles on the hydration of propargylic alcohols with IL-based systems with AgOAc and ZnO, respectively, showing a good knowledge of the underlying subject. This manuscript investigates the application of the CuCl-IL based system for the same purpose. The comprehensiveness of the manuscript is very good. The authors clearly state the aim of the work and briefly summarise developments in the field. The experiments and methods are very well described, and the obtained results properly discussed, including all the necessary spectra. The manuscript is well written, easy to read and comprehend, and meets the journal standards. In my opinion, this work is meaningful and can be published, after correcting small formatting errors.

Author Response

We appreciate the comments and suggestions of the reviewer. As the reviewer suggested, the manuscript was carefully checked again to correct the small formatting errors.

  1. The format in table 1 has been modified.
  2. The font and format in table 3 has been uniformed.

Reviewer 3 Report

Comments and Suggestions for Authors

The manuscript by Ye Yuan et al. describes a new route to synthesize α-hydroxy ketones. In particular the authors develop a CuCl/[C2C1im][Lev] system for the CO2-promoted hydration of propargylic alcohols under  CO2 pressure. This procedure, based on the use of a biomass based ionic liquid, avoids the use of organic volatile solvents or additives and provides high activity, wide substrate scope, and significant recyclability. The use of ionic liquids in synthesis is of large interest, mainly due to their environmental friendly properties.

A systematic study of the proposed synthesis process, focusing on the components and the conditions as well, allows the proposition of a mechanistic model which attributes to the ionic liquid a role both in the activation of the substates and in the generation of the adducts.

The paper is well organized and the experiments well planned. The results are clearly described and supported by experimental data. I think that the paper can be published as it is.

Just a couple of quick suggestion for the authors.

-In the “Materials and methods” section, the purity of the purchased components should be declared.

- In the “Proposed mechanism for the catalytic process” section, evidences supporting the formation of the transition state should be reported.

Comments on the Quality of English Language

The English is fine in my opinion and only minor editing of the language is required

Author Response

(The authors gave the same response as above.)

Reviewer 4 Report

Comments and Suggestions for Authors

The paper is detailed and interesting. However, I would suggest a schematic revision for the mechanism (Scheme 3 ) as it a bit confusing and not self-explanatory. For example, step 1 can be split into 1a showing the activation of the hydroxy group 1b) Cu co-ordination. 

Comments on the Quality of English Language

English language is fine except at few places the sentences are bit long. 

Author Response

(The authors gave the same response as above.)

Round 2

Reviewer 1 Report

Comments and Suggestions for Authors

The authors have addressed all the concerns. The manuscript is recommended for acceptance.